# Reconciling $\lambda$-Returns with Experience Replay

**Brett Daley**
Khoury College of Computer Sciences
Northeastern University
Boston, MA 02115
b.daley@northeastern.edu

**Christopher Amato**
Khoury College of Computer Sciences
Northeastern University
Boston, MA 02115
c.amato@northeastern.edu

## Abstract

Modern deep reinforcement learning methods have departed from the incremental learning required for eligibility traces, rendering the implementation of the $\lambda$-return difficult in this context. In particular, off-policy methods that utilize experience replay remain problematic because their random sampling of minibatches is not conducive to the efficient calculation of $\lambda$-returns. Yet replay-based methods are often the most sample efficient, and incorporating $\lambda$-returns into them is a viable way to achieve new state-of-the-art performance. Towards this, we propose the first method to enable practical use of $\lambda$-returns in arbitrary replay-based methods without relying on other forms of decorrelation such as asynchronous gradient updates. By promoting short sequences of past transitions into a small cache within the replay memory, adjacent $\lambda$-returns can be efficiently precomputed by sharing Q-values. Computation is not wasted on experiences that are never sampled, and stored $\lambda$-returns behave as stable temporal-difference (TD) targets that replace the target network. Additionally, our method grants the unique ability to observe TD errors prior to sampling; for the first time, transitions can be prioritized by their true significance rather than by a proxy to it. Furthermore, we propose the novel use of the TD error to dynamically select $\lambda$-values that facilitate faster learning. We show that these innovations can enhance the performance of DQN when playing Atari 2600 games, even under partial observability. While our work specifically focuses on $\lambda$-returns, these ideas are applicable to any multi-step return estimator.

## 1   Introduction

Eligibility traces [1, 15, 36] have been a historically successful approach to the credit assignment problem in reinforcement learning. By applying time-decaying 1-step updates to recently visited states, eligibility traces provide an efficient and online mechanism for generating the $\lambda$-return at each timestep [34]. The $\lambda$-return (equivalent to an exponential average of all $n$-step returns [39]) often yields faster empirical convergence by interpolating between low-variance temporal-difference (TD) returns [33] and low-bias Monte Carlo returns. Eligibility traces can be effective when the reward signal is sparse or the environment is partially observable.

More recently, deep reinforcement learning has shown promise on a variety of high-dimensional tasks such as Atari 2600 games [25], Go [32], 3D maze navigation [23], Doom [17], and robotic locomotion [6, 11, 18, 19, 29]. While neural networks are theoretically compatible with eligibility traces [34], training a non-linear function approximator online can cause divergence due to the strong correlations between temporally successive states [37]. Circumventing this issue has required unconventional solutions like experience replay [21], in which gradient updates are conducted using randomly sampled past experience to decorrelate the training data. Experience replay is also important for sample efficiency because environment transitions are reused multiple times rather than being discarded immediately. For this reason, well-tuned algorithms using experience replay such as

Rainbow [12] and ACER [38] are still among the most sample-efficient deep reinforcement learning methods today for playing Atari 2600 games.

The dependency of the $\lambda$-return on many future Q-values makes it prohibitively expensive to combine directly with minibatched experience replay when the Q-function is a deep neural network. Consequently, replay-based methods that use $\lambda$-returns (or derivative estimators like Retrace($\lambda$) [26]) have been limited to algorithms that can learn from long, sequential trajectories [8, 38] or utilize asynchronous parameter updates [24] to decorrelate such trajectories [26]. A general method for combining $\lambda$-returns with minibatch sampling would be useful for a vast array of off-policy algorithms including DQN [25], DRQN [10], SDQN [22], DDPG [20], NAF [7], and UVFA [30] that cannot learn from sequential trajectories like these.

In this paper, we present a general strategy for rectifying $\lambda$-returns and replayed minibatches of experience. We propose the use of a cache within the replay memory to store precomputed $\lambda$-returns and replace the function of a target network. The cache is formed from short sequences of experience that allow the $\lambda$-returns to be computed efficiently via recursion while maintaining an acceptably low degree of sampling bias. A unique benefit to this approach is that each transition's TD error can be observed before it is sampled, enabling novel sampling techniques that utilize this information. We explore these opportunities by prioritizing samples according to their actual TD error magnitude — rather than a proxy to it like in prior work [31] — and also dynamically selecting $\lambda$-values to facilitate faster learning. Together, these methods can significantly increase the sample efficiency of DQN when playing Atari 2600 games, even when the complete environment state is obscured. The ideas introduced here are general enough to be incorporated into any replay-based reinforcement learning method, where similar performance improvements would be expected.

## 2   Background

Reinforcement learning is the problem where an agent must interact with an unknown environment through trial-and-error in order to maximize its cumulative reward [34]. We first consider the standard setting where the environment can be formulated as a Markov Decision Process (MDP) defined by the 4-tuple $(\mathcal{S}, \mathcal{A}, \mathcal{P}, \mathcal{R})$. At a given timestep $t$, the environment exists in state $s_t \in \mathcal{S}$. The agent takes an action $a_t \in \mathcal{A}$ according to policy $\pi(a_t|s_t)$, causing the environment to transition to a new state $s_{t+1} \sim \mathcal{P}(s_t, a_t)$ and yield a reward $r_t \sim \mathcal{R}(s_t, a_t, s_{t+1})$. Hence, the agent's goal can be formalized as finding a policy that maximizes the expected discounted return $\mathbb{E}_\pi \left[ \sum_{i=0}^{H} \gamma^i r_i \right]$ up to some horizon $H$. The discount $\gamma \in [0, 1]$ affects the relative importance of future rewards and allows the sum to converge in the case where $H \to \infty, \gamma \neq 1$. An important property of the MDP is that every state $s \in \mathcal{S}$ satisfies the Markov property; that is, the agent needs to consider only the current state $s_t$ when selecting an action in order to perform optimally.

In reality, most problems of interest violate the Markov property. Information presently accessible to the agent may be incomplete or otherwise unreliable, and therefore is no longer a sufficient statistic for the environment's history [13]. We can extend our previous formulation to the more general case of the Partially Observable Markov Decision Process (POMDP) defined by the 6-tuple $(\mathcal{S}, \mathcal{A}, \mathcal{P}, \mathcal{R}, \Omega, \mathcal{O})$. At a given timestep $t$, the environment exists in state $s_t \in \mathcal{S}$ and reveals observation $o_t \sim \mathcal{O}(s_t), o_t \in \Omega$. The agent takes an action $a_t \in \mathcal{A}$ according to policy $\pi(a_t|o_0, \ldots, o_t)$ and receives a reward $r_t \sim \mathcal{R}(s_t, a_t, s_{t+1})$, causing the environment to transition to a new state $s_{t+1} \sim \mathcal{P}(s_t, a_t)$. In this setting, the agent may need to consider arbitrarily long sequences of past observations when selecting actions in order to perform well.[1]

We can mathematically unify MDPs and POMDPs by introducing the notion of an approximate state $\hat{s}_t = \phi(o_0, \ldots, o_t)$, where $\phi$ defines an arbitrary transformation of the observation history. In practice, $\phi$ might consider only a subset of the history — even just the most recent observation. This allows for the identical treatment of MDPs and POMDPs by generalizing the notion of a Bellman backup, and greatly simplifies our following discussion. However, it is important to emphasize that $\hat{s}_t \neq s_t$ in general, and that the choice of $\phi$ can affect the solution quality.

## 2.1 $\lambda$-returns

In the control setting, value-based reinforcement learning algorithms seek to produce an accurate estimate $Q(\hat{s}_t, a_t)$ of the expected discounted return achieved by following the optimal policy $\pi^*$ after taking action $a_t$ in state $\hat{s}_t$. Suppose the agent acts according to the (possibly suboptimal) policy $\mu$ and experiences the finite trajectory $\hat{s}_t, a_t, r_t, \hat{s}_{t+1}, a_{t+1}, r_{t+1}, \ldots, \hat{s}_T$. The estimate at time $t$ can be improved, for example, by using the $n$-step TD update [34]:

$$Q(\hat{s}_t, a_t) \leftarrow Q(\hat{s}_t, a_t) + \alpha \big[ R_t^{(n)} - Q(\hat{s}_t, a_t) \big] \tag{1}$$

where $R_t^{(n)}$ is the $n$-step return[2] and $\alpha$ is the learning rate controlling the magnitude of the update. When $n = 1$, Equation (1) is equivalent to Q-Learning [39]. In practice, the 1-step update suffers from slow credit assignment and high estimation bias. Increasing $n$ enhances the immediate sensitivity to future rewards and decreases the bias, but at the expense of greater variance which may require more samples to converge to the true expectation. Any valid return estimator can be substituted for the $n$-step return in Equation (1), including weighted averages of multiple $n$-step returns [34]. A popular choice is the $\lambda$-return, defined as the exponential average of every $n$-step return [39]:

$$R_t^\lambda = (1 - \lambda) \sum_{n=1}^{N-1} \lambda^{n-1} R_t^{(n)} + \lambda^{N-1} R_t^{(N)} \tag{2}$$

where $N = T - t$ and $\lambda \in [0, 1]$ is a hyperparameter that controls the decay rate. When $\lambda = 0$, Equation (2) reduces to the 1-step return. When $\lambda = 1$ and $\hat{s}_T$ is terminal, Equation (2) reduces to the Monte Carlo return. The $\lambda$-return can thus be seen a smooth interpolation between these methods.[3] When learning offline, it is often the case that a full sequence of $\lambda$-returns needs to be calculated. Computing Equation (2) repeatedly for each state in an $N$-step trajectory would require roughly $N + (N - 1) + \cdots + 1 = O(N^2)$ operations, which is impractical. Alternatively, given the full trajectory, the $\lambda$-returns can be calculated efficiently with recursion:

$$R_t^\lambda = R_t^{(1)} + \gamma \lambda \big[ R_{t+1}^\lambda - \max_{a' \in \mathcal{A}} Q(\hat{s}_{t+1}, a') \big] \tag{3}$$

We include a derivation in Appendix D for reference, but this formulation[4] has been commonly used in prior work [5, 27]. Because $R_t^\lambda$ can be computed given $R_{t+1}^\lambda$ in a constant number of operations, the entire sequence of $\lambda$-returns can be generated with $O(N)$ time complexity. Note that the $\lambda$-return presented here unconditionally conducts backups using the maximizing action for each $n$-step return, regardless of which actions were actually selected by the behavioral policy $\mu$. This is equivalent to Peng's Q($\lambda$) [27]. Although Peng's Q($\lambda$) has been shown to perform well empirically, its mixture of on- and off-policy data does not guarantee convergence [34]. One possible alternative is Watkin's Q($\lambda$) [39], which terminates the $\lambda$-return calculation by setting $\lambda = 0$ whenever an exploratory action is taken. Watkin's Q($\lambda$) provably converges to the optimal policy in the tabular case [26], but terminating the returns in this manner can slow learning [34].

## 2.2 Deep Q-Network

Deep Q-Network (DQN) was one of the first major successes of deep reinforcement learning, achieving human-level performance on Atari 2600 games using only the screen pixels as input [25]. DQN is the deep-learning analog of Q-Learning. Because maintaining tabular information for every state-action pair is not feasible for large state spaces, DQN instead learns a parameterized function $Q(\hat{s}_t, a_t; \theta)$ — implemented as a deep neural network — to generalize over states. Unfortunately, directly updating $Q$ according to a gradient-based version of Equation (1) does not work well [25, 37]. To overcome this, transitions $(\hat{s}_t, a_t, r_t, \hat{s}_{t+1})$ are stored in a replay memory $D$ and gradient descent is performed on uniformly sampled minibatches of past experience. A target network with stale parameters $\theta^-$ copied from $\theta$ every $F$ timesteps helps prevent oscillations of $Q$. Hence, DQN becomes a minimization problem where the following loss is iteratively approximated and reduced:

$$L(\theta) = \mathbb{E}_{(\hat{s},a,r,\hat{s}') \sim U(D)} \left[ \left( r + \gamma \max_{a' \in \mathcal{A}} Q(\hat{s}', a'; \theta^-) - Q(\hat{s}, a; \theta) \right)^2 \right]$$

DQN assumes Markovian inputs, but a single Atari 2600 game frame is partially observable. Hence, the four most-recent observations were concatenated together to form an approximate state in [25].

## 3 Experience replay with $\lambda$-returns

Deep reinforcement learning invariably utilizes offline learning schemes, making the recursive $\lambda$-return in Equation (3) ideal for these methods. Nevertheless, combining $\lambda$-returns with experience replay remains challenging. This is because the $\lambda$-return theoretically depends on all future Q-values. Calculating Q-values is notoriously expensive for deep reinforcement learning due to the neural network — an important distinction from tabular methods where Q-values can merely be retrieved from a look-up table. Even if the $\lambda$-return calculation were truncated after 10 timesteps, it would still require 10 times the computation of a 1-step method. This would be useful only in rare cases where maximal sample efficiency is desired at all costs.

An ideal $\lambda$-return algorithm using experience replay would more favorably balance computation and sample efficiency, while simultaneously allowing for arbitrary function approximators and learning methods. In this section, we propose several techniques to implement such an algorithm. For the purposes of our discussion, we use DQN to exemplify the ideas in the following sections, but they are applicable to any off-policy reinforcement learning method. We refer to this particular instantiation of our methods as DQN($\lambda$); the pseudocode is provided in Appendix B.

### 3.1 Refreshed $\lambda$-returns

Because the $\lambda$-return is substantially more expensive than the 1-step return, the ideal replay-based method minimizes the number of times each return estimate is computed. Hence, our principal modification of DQN is to store each return $R_t^\lambda$ along with its corresponding transition in the replay memory $D$. Training becomes a matter of sampling minibatches of precomputed $\lambda$-returns from $D$ and reducing the squared error. Of course, the calculation of $R_t^\lambda$ must be sufficiently deferred because of its dependency on future states and rewards; one choice might be to wait until a terminal state is reached and then transfer the episode's $\lambda$-returns to $D$. The new loss function becomes the following:

$$L(\theta) = \mathbb{E}_{(\hat{s},a,R^\lambda) \sim U(D)} \left[ \left( R^\lambda - Q(\hat{s}, a; \theta) \right)^2 \right]$$

There are two major advantages to this strategy. First, no computation is repeated when a transition is sampled more than once. Second, adjacent $\lambda$-returns in the replay memory can be calculated very efficiently with the recursive update in Equation (3). The latter point is crucial; while computing randomly accessed $\lambda$-returns may require 10 or more Q-values per $\lambda$-return as discussed earlier, computing them in reverse chronological order requires only one Q-value per $\lambda$-return.

One remaining challenge is that the stored $\lambda$-returns become outdated as the Q-function evolves, slowing learning when the replay memory is large. Fortunately, this presents an opportunity to eliminate the target network altogether. Rather than copying parameters $\theta$ to $\theta^-$ every $F$ timesteps, we *refresh* the $\lambda$-returns in the replay memory using the present Q-function. This achieves the same effect by providing stable TD targets, but eliminates the redundant target network.

### 3.2 Cache

Refreshing all of the $\lambda$-returns in the replay memory using the recursive formulation in Equation (3) achieves maximum Q-value efficiency by exploiting adjacency, and removes the need for a target network. However, this process is still prohibitively expensive for typical DQN implementations that have a replay memory capacity on the order of millions of transitions. To make the runtime invariant to the size of the replay memory, we propose a novel strategy where $\frac{S}{B}$ contiguous "blocks" of $B$ transitions are randomly promoted from the replay memory to build a cache $C$ of size $S$. By refreshing only this small memory and sampling minibatches directly from it, calculations are not wasted on $\lambda$-returns that are ultimately never used. Furthermore, each block can still be efficiently refreshed using Equation (3) as before. Every $F$ timesteps, the cache is regenerated from newly sampled blocks (Figure 1), once again obviating the need for a target network.

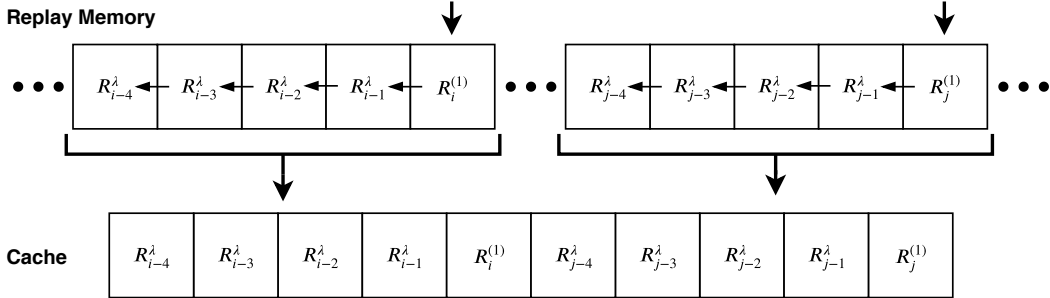

Figure 1: Our proposed cache-building process. For each randomly sampled index, a sequence ("block") of $\lambda$-returns is efficiently generated backwards via recursion. Together, the blocks form the new cache, which is treated as a surrogate for the replay memory for the following $F$ timesteps.

Caching is crucial to achieve practical runtime performance with $\lambda$-returns, but it introduces minor sample correlations that violate DQN's theoretical requirement of independently and identically distributed (*i.i.d.*) data. An important question to answer is how pernicious such correlations are in practice; if performance is not adversely affected — or, at the very least, the benefits of $\lambda$-returns overcome such effects — then we argue that the violation of the *i.i.d.* assumption is justified. In Figure 2, we compare cache-based DQN with standard target-network DQN on Seaquest and Space Invaders using $n$-step returns (all experimental procedures are detailed later in Section 5). Although the sampling bias of the cache decreases performance on Seaquest, the loss can be mostly recovered by increasing the cache size $S$. On the other hand, Space Invaders provides an example of a game where the cache actually outperforms the target network despite this bias. In our later experiments, we find that the choice of the return estimator has a significantly larger impact on performance than these sampling correlations do, and therefore the bias matters little in practice.

### 3.3 Directly prioritized replay

To our knowledge, DQN($\lambda$) is the first method with experience replay to compute returns before they are sampled, meaning it is possible to observe the TD errors of transitions prior to replaying them. This allows for the opportunity to specifically select samples that will facilitate the fastest learning. While prioritized experience replay has been explored in prior work [31], these techniques rely on the previously seen (and therefore outdated) TD error as a proxy for ranking samples. This is because the standard target-network approach to DQN computes TD errors as transitions are sampled, only to immediately render them inaccurate by the subsequent gradient update. Hence, we call our approach *directly* prioritized replay to emphasize that the true TD error is initially used. The tradeoff of our method is that only samples within the cache — not the full replay memory — can be prioritized.

While any prioritization distribution is possible, we propose a mixture between a uniform distribution over $C$ and a uniform distribution over the samples in $C$ whose absolute TD errors exceed some quantile. An interesting case arises when the chosen quantile is the median; the distribution becomes symmetric and has a simple analytic form. Letting $p \in [0, 1]$ be our interpolation hyperparameter and $\delta_i$ represent the (unique) error of sample $x_i \in C$, we can write the sampling probability explicitly:

$$P(x_i) = \begin{cases} \frac{1}{S}(1 + p) & \text{if } |\delta_i| > \text{median}(|\delta_0|, |\delta_1|, \dots, |\delta_{S-1}|) \\ \frac{1}{S} & \text{if } |\delta_i| = \text{median}(|\delta_0|, |\delta_1|, \dots, |\delta_{S-1}|) \\ \frac{1}{S}(1 - p) & \text{if } |\delta_i| < \text{median}(|\delta_0|, |\delta_1|, \dots, |\delta_{S-1}|) \end{cases}$$

A distribution of this form is appealing because it is scale-invariant and insensitive to noisy TD errors, helping it to perform consistently across a wide variety of reward functions. Following previous work [31], we linearly anneal $p$ to 0 during training to alleviate the bias caused by prioritization.

### 3.4 Dynamic $\lambda$ selection

The ability to analyze $\lambda$-returns offline presents a unique opportunity to dynamically choose $\lambda$-values according to certain criteria. In previous work, tabular reinforcement learning methods have

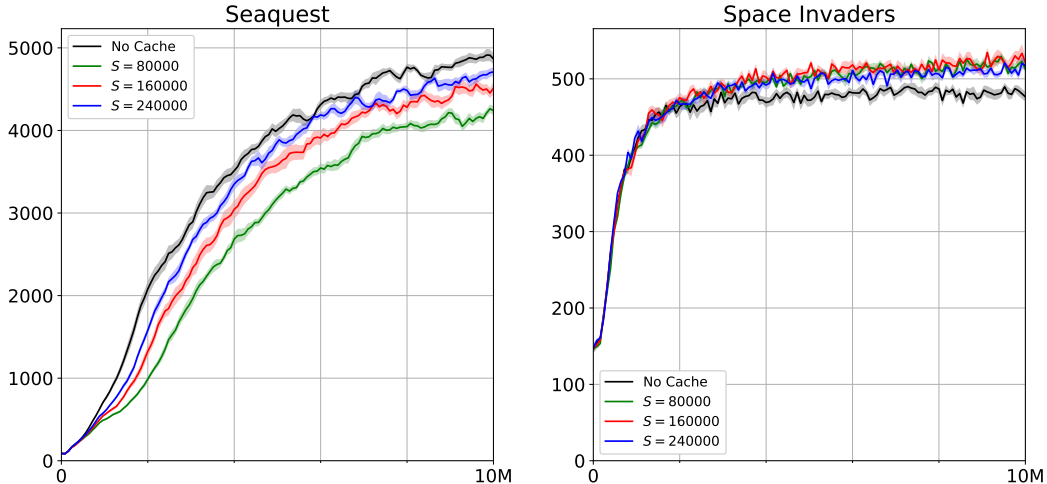

Figure 2: Ablation analysis of our caching method on Seaquest and Space Invaders. Using the 3-step return with DQN for all experiments, we compared the scores obtained by caches of size $S \in \{80000, 160000, 240000\}$ against a target-network baseline. As expected, the cache's violation of the *i.i.d.* assumption has a negative performance impact on Seaquest, but this can be mostly recovered by increasing $S$. Surprisingly, the trend is reversed for Space Invaders, indicating that the cache's sample correlations do not always harm performance. Because the target network is impractical for computing $\lambda$-returns, the cache is effective when $\lambda$-returns outperform $n$-step returns.

utilized variable $\lambda$-values to adjust credit assignment according to the number of times a state has been visited [34, 35] or a model of the $n$-step return variance [16]. In our setting, where function approximation is used to generalize across a high-dimensional state space, it is difficult to track state-visitation frequencies and states might not be visited more than once. Alternatively, we propose to select $\lambda$-values based on their TD errors. One strategy we found to work well empirically is to compute several different $\lambda$-returns and then select the median return at each timestep. Formally, we redefine $R_t^\lambda = \mathrm{median}(R_t^{\lambda=0/k}, R_t^{\lambda=1/k}, \dots, R_t^{\lambda=k/k})$, where $k + 1$ is the number of evenly spaced candidate $\lambda$-values. We used $k = 20$ for all of our experiments; larger values yielded marginal benefit. Median-based selection is appealing because it integrates multiple $\lambda$-values in an intuitive way and is robust to outliers that could cause destructive gradient updates. In Appendix C, we also experimented with selecting $\lambda$-values that bound the mean absolute error of each cache block, but we found median-based selection to work better in practice.

## 4 Related work

The $\lambda$-return has been used in prior work to improve the sample efficiency of Deep Recurrent Q-Network (DRQN) for Atari 2600 games [8]. Because recurrent neural networks (RNNs) produce a sequence of Q-values during truncated backpropagation through time, these precomputed values can be exploited to calculate $\lambda$-returns with little additional expense over standard DRQN. The problem with this approach is its lack of generality; the Q-function is restricted to RNNs, and the length $N$ over which the $\lambda$-return is computed must be constrained to the length of the training sequence. Consequently, increasing $N$ to improve credit assignment forces the training sequence length to be increased as well, introducing undesirable side effects like exploding and vanishing gradients [3] and a substantial runtime cost. Additionally, the use of a target network means $\lambda$-returns must be recalculated on every training step, even when the input sequence and Q-function do not change. In contrast, our proposed caching mechanism only periodically updates stored $\lambda$-returns, thereby avoiding repeated calculations and eliminating the need for a target network altogether. This strategy provides maximal flexibility by decoupling the training sequence length from the $\lambda$-return length and makes no assumptions about the function approximator. This allows it to be incorporated into any replay-based algorithm and not just DRQN.

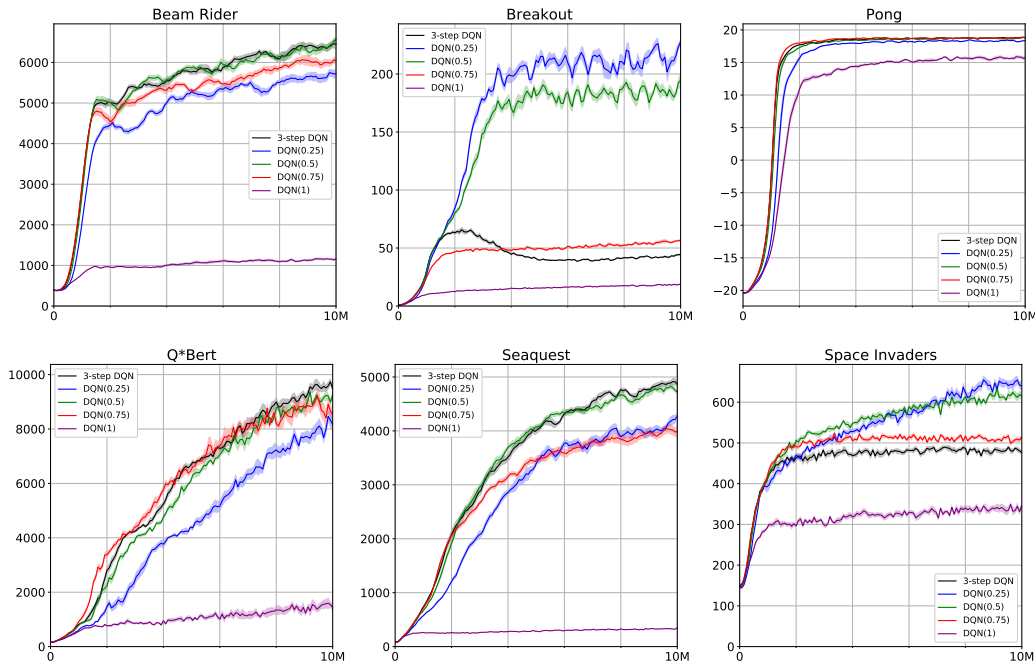

Figure 3: Sample efficiency comparison of DQN($\lambda$) with $\lambda \in \{0.25, 0.5, 0.75, 1\}$ against 3-step DQN on six Atari games.

## 5 Experiments

In order to characterize the performance of DQN($\lambda$), we conducted numerous experiments on six Atari 2600 games. We used the OpenAI Gym [4] to provide an interface to the Arcade Learning Environment [2], where observations consisted of the raw frame pixels. We compared DQN($\lambda$) against a standard target-network implementation of DQN using the 3-step return, which was shown to work well in [12]. We matched the hyperparameters and procedures in [25], except we trained the neural networks with Adam [14]. Unless stated otherwise, $\lambda$-returns were formulated as Peng's Q($\lambda$).

For all experiments in this paper, agents were trained for 10 million timesteps. An agent's performance at a given time was evaluated by averaging the earned scores of its past 100 completed episodes. Each experiment was averaged over 10 random seeds with the standard error of the mean indicated. Our complete experimental setup is discussed in Appendix A.

**Peng's Q($\lambda$):** We compared DQN($\lambda$) using Peng's Q($\lambda$) for $\lambda \in \{0.25, 0.5, 0.75, 1\}$ against the baseline on each of the six Atari games (Figure 3). For every game, at least one $\lambda$-value matched or outperformed the 3-step return. Notably, $\lambda \in \{0.25, 0.5\}$ yielded huge performance gains over the baseline on Breakout and Space Invaders. This finding is quite interesting because $n$-step returns have been shown to perform poorly on Breakout [12], suggesting that $\lambda$-returns can be a better alternative.

**Watkin's Q($\lambda$):** Because Peng's Q($\lambda$) is a biased return estimator, we repeated the previous experiments using Watkin's Q($\lambda$). The results are included in Appendix E. Surprisingly, Watkin's Q($\lambda$) failed to outperform Peng's Q($\lambda$) on every environment we tested. The worse performance is likely due to the cut traces, which slow credit assignment in spite of their bias correction.

**Directly prioritized replay and dynamic $\lambda$ selection:** We tested DQN($\lambda$) with prioritization $p = 0.1$ and median-based $\lambda$ selection on the six Atari games. The results are shown in Figure 4. In general, we found that dynamic $\lambda$ selection did not improve performance over the best hand-picked $\lambda$-value; however, it always matched or outperformed the 3-step baseline without any manual $\lambda$ tuning.

**Partial observability:** In Appendix F, we repeated the experiments in Figure 4 but provided agents with only a 1-frame input to make the environments partially observable. We hypothesized that the relative performance difference between DQN($\lambda$) and the baseline would be greater under partial observability, but we found that it was largely unchanged.

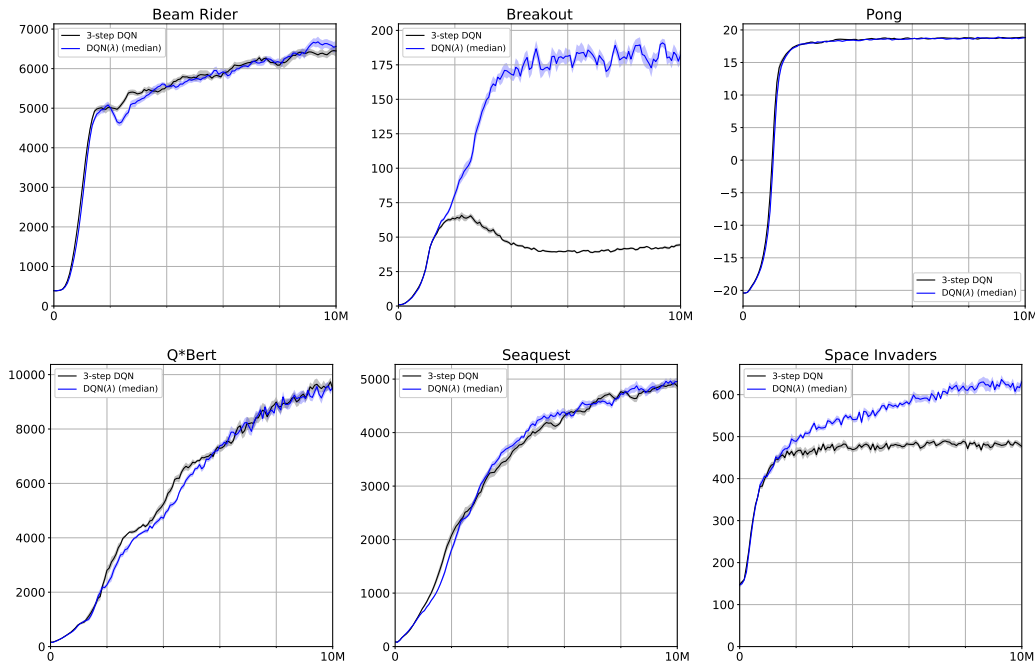

Figure 4: Sample efficiency comparison of DQN($\lambda$) with prioritization $p = 0.1$ and median-based dynamic $\lambda$ selection against 3-step DQN on six Atari games.

**Real-time sample efficiency:** In certain scenarios, it may be desirable to train a model as quickly as possible without regard to the number of environment samples. For the best $\lambda$-value we tested on each game in Figure 3, we plotted the score as a function of wall-clock time and compared it against the target-network baseline in Appendix G. Significantly, DQN($\lambda$) completed training faster than DQN on five of the six games. This shows that the cache can be more computationally efficient than a target network. We believe the speedup is attributed to greater GPU parallelization when computing Q-values because the cache blocks are larger than a typical minibatch.

## 6 Conclusion

We proposed a novel technique that allows for the efficient integration of $\lambda$-returns into any off-policy method with minibatched experience replay. By storing $\lambda$-returns in a periodically refreshed cache, we eliminate the need for a target network and enable offline analysis of the TD errors prior to sampling. This latter feature is particularly important, making our method the first to directly prioritize samples according to their actual loss contribution. To our knowledge, our method is also the first to explore dynamically selected $\lambda$-values for deep reinforcement learning. Our experiments showed that these contributions can increase the sample efficiency of DQN by a large margin.

While our work focused specifically on $\lambda$-returns, our proposed methods are equally applicable to any multi-step return estimator. One avenue for future work is to utilize a lower-variance, bias-corrected return such as Tree Backup [28], Q*($\lambda$) [9], or Retrace($\lambda$) [26] for potentially better performance. Furthermore, although our method does not require asynchronous gradient updates, a multi-threaded implementation of DQN($\lambda$) could feasibly enhance both absolute and real-time sample efficiencies. Our ideas presented here should prove useful to a wide range of off-policy reinforcement learning methods by improving performance while limiting training duration.

### Acknowledgments

We would like to thank the anonymous reviewers for their valuable feedback. We also gratefully acknowledge NVIDIA Corporation for its GPU donation. This research was funded by NSF award 1734497 and an Amazon Research Award (ARA).

## Footnotes

[1] To achieve optimality, the policy must additionally consider the action history in general.

[2] Defined as $R_t^{(n)} = r_t + \gamma r_{t+1} + \cdots + \gamma^{n-1} r_{t+n-1} + \gamma^n \max_{a' \in \mathcal{A}} Q(\hat{s}_{t+n}, a')$, $n \in \{1, 2, \ldots, T - t\}$.

[3] Additionally, the monotonically decreasing weights can be interpreted as the recency heuristic, which assumes that recent states and actions are likelier to have contributed to a given reward [34].

[4] The equation is sometimes rewritten: $R_t^\lambda = r_t + \gamma \big[ \lambda R_{t+1}^\lambda + (1 - \lambda) \max_{a' \in \mathcal{A}} Q(\hat{s}_{t+1}, a') \big]$.

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
