[Supplementary Material]

# A Experimental setup

All game frames were subjected to the same preprocessing steps described in [25]. We converted images to grayscale and downsized them to $84 \times 84$ pixels. Rewards were clipped to $\{-1, 0, +1\}$. For equal comparison, we used the same convolutional neural network from [25] for all agents: three convolutional layers followed by two fully connected layers. During training, $\epsilon$-greedy exploration was linearly annealed from 1 to 0.1 over the first one million timesteps and then held constant. We trained the neural networks using Adam [14] with $\alpha = 10^{-4}$, $\beta_1 = 0.9$, $\beta_2 = 0.999$, and $\epsilon = 10^{-4}$.

Our chosen hyperparameters are shown in Table 1. The table is divided into two portions; the upper section contains hyperparameters that are identical to those in [25] (although possibly denoted by a different symbol), while the lower section contains new hyperparameters introduced in our work.

Table 1: Hyperparameters for DQN($\lambda$)

| Hyperparameter | Symbol | Value | Description |
|---|---|---|---|
| minibatch size | M | 32 | Number of samples used to compute a single gradient descent step. |
| replay memory size | | 1000000 | Maximum number of samples that can be stored in the replay memory before old samples are discarded. |
| agent history length | | 4 | Number of recent observations (game frames) simultaneously fed as input to the neural network. |
| refresh frequency | $F$ | 10000 | Frequency, measured in timesteps, at which the target network is updated for DQN or the cache is rebuilt for DQN($\lambda$). |
| discount factor | $\gamma$ | 0.99 | Weighting coefficient that influences the importance of future rewards. |
| replay start size | $N$ | 50000 | Number of timesteps for which to initially execute a uniform random policy and pre-populate the replay memory. |
| cache size | $S$ | 80000 | Number of samples used to build the cache upon refresh. |
| block size | $B$ | 100 | Atomic length of the sampled sequences that are promoted into the cache upon refresh. |

When sizing the cache for DQN($\lambda$), we made sure that the number of minibatches per timestep (1:4 ratio) was preserved for fair comparison. Specifically,

$$\frac{10000 \, \text{timesteps}}{1 \, \text{refresh}} \times \frac{1 \, \text{minibatch}}{4 \, \text{timesteps}} \times \frac{32 \, \text{samples}}{1 \, \text{minibatch}} = \frac{80000 \, \text{samples}}{1 \, \text{refresh}}$$

Hence, we used $S = 80000$ for all experiments. To help reduce bias when building the cache, we permitted overlapping blocks. That is, we did not check for boundary constraints of nearby blocks, nor did we align blocks to a fixed grid (in contrast to a real CPU cache). This meant that multiple copies of the same experience might have existed in the cache simultaneously, but each experience therefore had the same probability of being promoted.

# B Algorithm

The standard implementation of DQN($\lambda$) is given below in Algorithm 1. For clarity, enhancements such as prioritized experience replay and dynamic $\lambda$ selection are not shown.

---

**Algorithm 1** DQN($\lambda$)

---

**function** BUILD-CACHE($D$)
    Initialize empty list $C$
    **for** $1, 2, \ldots, \frac{S}{B}$ **do**
        Sample block $(\hat{s}_k, a_k, r_k, \hat{s}_{k+1}), \ldots, (\hat{s}_{k+B-1}, a_{k+B-1}, r_{k+B-1}, \hat{s}_{k+B})$ from $D$
        $R^\lambda \leftarrow \max_{a' \in \mathcal{A}} Q(\hat{s}_{k+B}, a'; \theta)$
        **for** $i \in \{k + B - 1, k + B - 2, \ldots, k\}$ **do**
$$R^\lambda \leftarrow \begin{cases} r_i & \text{if terminal}(\hat{s}_{i+1}) \\ r_i + \gamma[\lambda R^\lambda + (1 - \lambda)\max_{a' \in \mathcal{A}} Q(\hat{s}_{i+1}, a'; \theta)] & \text{otherwise} \end{cases}$$
            Append tuple $(\hat{s}_i, a_i, R^\lambda)$ to $C$
        **end for**
    **end for**
    **return** $C$
**end function**

Initialize replay memory $D$ with $N$ experiences
Initialize parameter vector $\theta$ randomly
Initialize state $\hat{s}_0 = \phi(o_0)$

**for** $t \in \{0, 1, \ldots, T - 1\}$ **do**
    **if** $t \equiv 0 \bmod F$ **then**
        $C \leftarrow$ BUILD-CACHE($D$)
        **for** $1, 2, \ldots, \frac{S}{M}$ **do**
            Sample minibatch $(\hat{s}_j, a_j, R^\lambda_j)$ from $C$
            Perform gradient descent step on $\left[R^\lambda_j - Q(\hat{s}_j, a_j; \theta)\right]^2$ with respect to $\theta$
        **end for**
    **end if**
    Execute $a_t = \begin{cases} a \sim U(\mathcal{A}) & \text{with probability } \epsilon \\ \text{argmax}_{a' \in \mathcal{A}} Q(\hat{s}_t, a'; \theta) & \text{otherwise} \end{cases}$
    Receive reward $r_t$ and new observation $o_{t+1}$
    Approximate state $\hat{s}_{t+1} = \phi(o_0, \ldots, o_{t+1})$
    Store transition $(\hat{s}_t, a_t, r_t, \hat{s}_{t+1})$ in $D$
**end for**

---

# C Dynamic $\lambda$ selection with bounded TD error

We experimented with an alternative dynamic $\lambda$ selection method that bounds the mean absolute TD error of each block when refreshing the cache. The squared error loss of DQN is known to be susceptible to large and potentially destructive gradients; this originally motivated reward clipping when playing Atari 2600 games in [25]. We hypothesized that bounding the error with dynamically selected $\lambda$-values would help prevent learning instability and improve sample efficiency.

Let $\bar{\delta}$ be our error bound (a hyperparameter), and let $L(\lambda) = \frac{1}{B} \sum_{i=0}^{B-1} |R_i^\lambda - Q(\hat{s}_i, a_i; \theta)|$ be the mean absolute TD error of the cache block being refreshed. Assuming that $L(\lambda)$ increases monotonically with $\lambda$, our target $\lambda$-value can be defined as the following:

$$\lambda^* = \underset{\lambda}{\mathrm{argmax}}\, L(\lambda) \quad \text{subject to} \quad L(\lambda) \leq \bar{\delta} \quad \text{and} \quad 0 \leq \lambda \leq 1$$

In practice, to efficiently find a $\lambda$-value that approximately solves this equation, we conducted a binary search with a maximum depth of 7. We also tested the extreme $\lambda$-values (*i.e.* $\lambda = 0$ and $\lambda = 1$) prior to the binary search because we found that these often exceeded or satisfied the $\bar{\delta}$ constraint, respectively. This allowed our procedure to sometimes return early and reduce the average runtime.

The results are given in Figure 5. While the final scores on Breakout and Space Invaders were improved over the 3-step baseline, performance also decreased significantly on Beam Rider, Q*Bert, and Seaquest. For this reason, we recommend the use of median-based dynamic $\lambda$ selection instead.

Figure 5: Sample efficiency comparison of DQN($\lambda$) with prioritization $p = 0.1$ and error-based dynamic $\lambda$ selection against 3-step DQN on six Atari games. The mean absolute TD error of each block was roughly bounded by $\bar{\delta} = 0.025$ during the refresh procedure.

# D  Derivation of recursive $\lambda$-return calculation

Suppose the agent experiences the finite trajectory $\hat{s}_t, a_t, r_t, \ldots, \hat{s}_{T-1}, a_{T-1}, r_{T-1}, \hat{s}_T$. We wish to write $R_t^\lambda$ as a function of $R_{t+1}^\lambda$. First, note the general recursive relationship between $n$-step returns:

$$R_k^{(n)} = r_k + \gamma R_{k+1}^{(n-1)} \tag{4}$$

Let $N = T - t$. Starting with the definition of the $\lambda$-return,

$$R_t^\lambda = \left[ (1-\lambda) \sum_{n=1}^{N-1} \lambda^{n-1} R_t^{(n)} + \lambda^{N-1} R_t^{(N)} \right]$$

$$= (1-\lambda) R_t^{(1)} + \left[ (1-\lambda) \sum_{n=2}^{N-1} \lambda^{n-1} R_t^{(n)} + \lambda^{N-1} R_t^{(N)} \right]$$

$$= (1-\lambda) R_t^{(1)} + \left[ (1-\lambda) \sum_{n=2}^{N-1} \lambda^{n-1} \left( r_t + \gamma R_{t+1}^{(n-1)} \right) + \lambda^{N-1} \left( r_t + \gamma R_{t+1}^{(N-1)} \right) \right] \tag{5}$$

$$= (1-\lambda) R_t^{(1)} + \lambda r_t + \gamma \lambda \left[ (1-\lambda) \sum_{n=2}^{N-1} \lambda^{n-2} R_{t+1}^{(n-1)} + \lambda^{N-2} R_{t+1}^{(N-1)} \right] \tag{6}$$

$$= (1-\lambda) R_t^{(1)} + \lambda r_t + \gamma \lambda \left[ (1-\lambda) \sum_{n'=1}^{N-2} \lambda^{n'-1} R_{t+1}^{(n')} + \lambda^{N-2} R_{t+1}^{(N-1)} \right] \tag{7}$$

$$= (1-\lambda) R_t^{(1)} + \lambda r_t + \gamma \lambda R_{t+1}^\lambda$$

$$= R_t^{(1)} - \lambda R_t^{(1)} + \lambda r_t + \gamma \lambda R_{t+1}^\lambda$$

$$= R_t^{(1)} - \lambda \left( r_t + \gamma \max_{a' \in \mathcal{A}} Q(\hat{s}_{t+1}, a') \right) + \lambda r_t + \gamma \lambda R_{t+1}^\lambda$$

$$= R_t^{(1)} + \gamma \lambda \left[ R_{t+1}^\lambda - \max_{a' \in \mathcal{A}} Q(\hat{s}_{t+1}, a') \right] \tag{8}$$

Equation (5) follows from the recursive relationship in Equation (4). Equation (6) follows from the telescoping identity $\lambda = (1-\lambda)\lambda + (1-\lambda)\lambda^2 + \cdots + (1-\lambda)\lambda^{N-2} + \lambda^{N-1}$. In Equation (7), we let $n' = n - 1$ to obtain an expression for $R_{t+1}^\lambda$. A sequence of $\lambda$-returns can be generated efficiently by repeatedly applying Equation (8). The recursion can be initialized using the fact that $R_T^\lambda = \max_{a' \in \mathcal{A}} Q(\hat{s}_T, a')$. If $\hat{s}_T$ is terminal, then $\max_{a' \in \mathcal{A}} Q(\hat{s}_T, a') = 0$ by definition.

# E   Watkin's Q($\lambda$)

Watkin's Q($\lambda$) is the simplest form of bias correction for Q-Learning with $\lambda$-returns. Whenever a non-greedy action is taken, $\lambda$ is set to 0, effectively performing a standard 1-step backup at the current timestep. This is apparent from substituting $\lambda = 0$ into Equation (3).

Watkin's Q($\lambda$) ensures that backups are conducted using greedy actions with respect to the Q-function, and therefore all updates are on-policy. In theory, we would expect this to improve performance by increasing the accuracy of return estimation, but our experiments in Figure 6 indicated otherwise. Although cutting traces to correct bias is well motivated, the overall performance was still significantly hampered compared to Peng's Q($\lambda$) (which ignores bias correction). This suggests that long-term credit assignment is more important than bias correction for the six games we tested.

Figure 6:   Comparison of DQN($\lambda$) with $\lambda \in \{0.25, 0.5, 0.75, 1\}$ against 3-step DQN on six Atari games. The $\lambda$-returns were formulated as Watkin's Q($\lambda$).

# F    Partial observability

Atari 2600 games are partially observable given a single game frame because the velocities of moving objects cannot be deduced. We repeated our median-based dynamic $\lambda$ experiments under this condition (Figure 7). In this case, $\lambda$-returns can theoretically help resolve state uncertainty by integrating information across multiple $n$-step returns. The agents performed worse, as expected, but the relative results were qualitatively similar to those of the fully observed experiments.

Figure 7: Sample efficiency comparison of DQN($\lambda$) with prioritization $p = 0.1$ and median-based dynamic $\lambda$ selection against 3-step DQN on six Atari games. Only a single frame at a time was presented to the agent to make the games partially observable.

# G   Real-time sample efficiency

We compared DQN($\lambda$) using each game's best tested $\lambda$-value against the 3-step DQN baseline and plotted the results as a function of elapsed time in hours (Figure 8). DQN($\lambda$) and its cache-based sampling method completed training faster than DQN on five of the six games, likely because the block size $B = 100$ is larger than the minibatch size $M = 32$ and can be more efficiently executed on a GPU.

Figure 8:    Real-time sample efficiency comparison of DQN($\lambda$) with the best $\lambda$-value in $\{0.25, 0.5, 0.75, 1\}$ against 3-step DQN on six Atari games. The horizontal axis indicates wall-clock time in hours.