[Reviews · NeurIPS 2019]

Reviewer 1



Update after rebuttal: I thank the authors for addressing the main points that I brought up in my original review. In particular, I believe that they did a good job in making some of the intuitions underlying their design choices clearer; including those clarifications in the text would make the paper's claims more easily understood by the readers. Overall, I also believe that the rebuttal did a good job at explaining the high-level rationale underlying the proposed method, even though I still feel like the technique doesn't always comes with a solid theory behind it. On the other hand, as R3 mentions, standard TD(lambda) doesn't have particularly solid theory surrounding it either, and that needs to be taken into account when judging how solid the contributions of this paper are. I have kept my original scores for this submission. -------------------- This paper introduces a technique to allow for the practical use of lambda-returns in combination with replay-based methods. This is an important problem because, while replay methods are important to decorrelate samples, they typically rely on sampling random minibatches of previous experiences, which makes it hard to readily adapt them to perform the calculations required by lambda-return methods. The paper proposes to store short sequences of past experiences into a cache data structure within the replay memory so that adjacent lambda-returns can be rapidly precomputed. Moreover, the stored lambda-returns can be interpreted as stable versions of target TD errors, which allows the authors to build a training architecture that does not rely on the typical target network used in Deep RL. This is an interesting paper and (as far as I can tell) it does introduce an original contribution to the field. It is, overall, clear and accessible---except for a few parts of the text that I mention below. I have a few comments and questions: 1) in Section 3.1, the authors argue that computing all of the lambda-returns in the replay memory requires only one Q-value per lambda-return, on average. This is not immediately clear to me. Could you please clarify why this is this case? 2) while I understand that eliminating a target network is an interesting and valuable contribution, the proposed solution requires "refreshing" all lambda-returns in the replay memory every F steps. In practice, other than requiring less memory, what is the main advantage of this method, vs. having to replace the current network with target network every F steps? 3) I do not fully understand the theoretical justification (and implications) of your decision to construct a cache of experience chunks by mixing one copy of the cache that is randomly shuffled and another one that is sorted by TD errors. Why is this needed and what happens if you use data originating from just one type of cache; i.e, only data that is completely random, or just data that sorted by its TD error? 4) regarding Figure 2, it is not clear what is being shown here. What are the x and y axes showing? Also, all curves look extremely similar: what is the conclusion that one should take from this analysis? 5) I do not fully understand the theoretical implications of the oversampling ratio hyperparameter (X) that you propose. This seems like a design decision that is not principled but that may work well in practice. Could you please discuss in which (more general) situations this approach would be appropriate? 6) the authors argue that their approach for efficiently computing lamba-returns is possible while maintaining an acceptably low degree of sampling bias. This is achieved by using a prioritization hyperparameter p, which needs to be properly annealed over time. Could you please discuss the implications of the annealing process (e.g. how fast p goes down to zero) in the transient properties of the trained network? I.e., *during* training, what happens when p is large (close to 1) and what could the negative effects be if we anneal it (say) too slowly or too quickly?

Reviewer 2



I've read the other reviews and authors' rebuttal. As the other reviewers, I do appreciate the proposed framework which enables the integration of lambda-return and experience replay at reasonable cost. In addition, the authors’ response seemingly addresses the other reviewers’ concerns on the design rationales. Hence, I’ve increased my assessment on the overall merit by one point, although I still have some concerns that the authors may present only favorable results for the specific choice of new parameters on the main idea of cashing. The authors’ rebuttal provides some useful intuitions on how to tune one of the hyper-parameters, X, but I was expecting concrete justification on the choice of hyper-parameters that are newly introduced in this paper, or study on the impact of each of new parameters. Nevertheless, as R1&3 mentioned, it may be beneficial to widen the spectrum of available/doable RL algorithms for the community, so I’m leaning towards accepting this paper. ---- The presentation is clear and well-organized. However, it is unfortunate that there is no study on the computational complexity of the proposed caching scheme, although reducing computational cost seems the main contribution. In addition, in the experiments, the gain from using lambda-returns compared to the standard 3-step DQN is inconsistent and even negative in some cases. Therefore, it is hard to believe that the proposed framework indeed provides some performance improvement other than gain from just introducing new parameters (e.g., X and lambda) to be optimized.

Reviewer 3



Update after rebuttal: My original score was primarily based on the core idea, which seems like a reasonable way to integrate lambda and experience replay, but I also agree with the concerns of the other reviewers. As for the seemingly ad-hoc nature of some of the design choices, the rebuttal did a reasonable job at explaining the high-level rationale, but it didn't provide solid theory to justify these choices. At the same time, standard TD(lambda) doesn't have particularly solid theory surrounding it either, but is often useful in practice. I also appreciate the addition of the median experiment, which in practice seems to do a bit better than dynamic lambda, but also feels a bit ad hoc. One concerning thing is that I strongly disagree with one point made in the rebuttal -- I don't think that it can be argued that Fig 8 says anything coherent about the relative computational cost in general, given the extreme variance in performance across problems. On a related note, I also have concerns that the paper does not establish that the variance across problems is due to something fundamental about lambda, rather than an artifact of one of the design choices. Overall, I've dropped my score two points after considering the points that the other reviewers made and taking the rebuttal into account. -------------- This paper is significantly original, in that it comes up with clever ways to apply old ideas (eligibility traces) to a new setting (deep RL with experience replay) in an efficient way, where others have failed. The methods are technically sound, and even the slightly hack-y elements (such as the internal cache and oversampling) are well-motivated and there is sufficient discussion of the issues that they can cause (e.g. slight sampling bias from the cache). The paper is very well written, clear, and significant for reasons stated above. Now, a few criticisms and/or opportunities for improvement: - In the Peng's Q(lambda) experiements, the paper points out that at least one value of lambda matched or outperformed the baseline 3-step return. However, it is a different value of lambda for almost every problem, and many of the values do much worse than the baseline in some problems. Since there is no clear way to set lambda (the dynamic settings aren't reliable either), this adds another hyperparamter to tune. However, this has always been a problem of lambda-based methods, and has not stopped the field from using them. I still think the core contribution is solid and that future work can maybe figure out these issues. - However, given the weakness of the results, it seems overly optimistic for the paper to claim that "Our experiments showed that these contributions improve the sample efficiency of DQN by a large factor". This claim should be scaled back and made much more conditional. - When discussing auto-tunin / dynamically selecting lambda, one piece of literature worth citing might be work that side-stepped the problem of setting lambda by formulating a parameter-free way of doing eligibility traces: @inproceedings{konidaris2011td_gamma, title={TD\_gamma: Re-evaluating Complex Backups in Temporal Difference Learning}, author={Konidaris, George and Niekum, Scott and Thomas, Philip S}, booktitle={Advances in Neural Information Processing Systems}, pages={2402--2410}, year={2011} }

[Author Response · NeurIPS 2019]

Thank you for the feedback. We presented an efficient method to combine $\lambda$-returns and experience replay for the first
time, yielding faster learning on several Atari games. This method is generally applicable to replay-based learning and
enables interesting avenues for prioritized replay and dynamic $\lambda$ selection, both of which we explored in our work.

Addressing **Reviewer 3's** comments, dynamic $\lambda$ selection performed well on only some of the games. We have explored
an alternative strategy (see figure below) that outperforms the baseline on 3 games and performs comparably on the
other 3. We define $R_t^\lambda = \text{median}(R_t^{\lambda=0/k}, R_t^{\lambda=1/k}, ..., R_t^{\lambda=k/k})$, where we used $k = 20$. That is, we compute many $\lambda$-returns
offline, and then select the median at each timestep. This is appealing because it integrates multiple $\lambda$-values in an
intuitive way, is robust to outliers, and eliminates the hyperparameter $\delta$. Because it does not require a binary search, the
expected runtime is similar or better. We can include these results in the main paper or the supplementary material.

**Reviewer 2** expressed concerns about computational cost. Figure 8 (supplementary material) shows that DQN($\lambda$) runs
about 33% faster than standard DQN. This is because cache chunks are larger than a typical minibatch, so we achieve
much better GPU parallelization when training. We can include additional runtime figures in the final paper, if desired.

**Reviewer 1** asked 6 questions, which we answer here:

**1.** Each $\lambda$-return needs only 1 Q-value due to the recursive formula: $R_t^\lambda = R_t^{(1)} + \gamma\lambda[R_{t+1}^\lambda - \max_{a' \in \mathcal{A}} Q(\hat{s}_{t+1}, a')]$. Conse-
quently, a 100-length chunk will require 100 Q-values, *i.e.* an average of 1 Q-value per $\lambda$-return.

**2.** The main practical benefit of refreshing $Q$-values (instead of using a target network) is saving memory.

**3 & 6.** We prioritize experience by stochastically mixing two *truncated* cache copies (one sorted by TD error magnitude,
one shuffled) with probability parameter $p$. The shuffling/sorting operations occur before truncation; therefore, on
average, the sorted copy has a higher absolute TD error than the shuffled copy, but it is also biased. Training solely
on the sorted copy ($p = 1$) could yield a poor policy as a result. On the other hand, $p = 0$ would completely disable
prioritization and could miss opportunities for faster learning.

Consequently, we recommend annealing $p \to 0$ to initially encourage faster learning and later facilitate accurate Q-value
estimation. Annealing $p$ too slowly could result in convergence to a bad policy. Annealing $p$ too quickly could slow
learning. While we experimented with this new technique, prioritization methods from previous work are certainly
possible here too, and would benefit from the true TD error information provided by the cache.

**4.** Figure 2 is our ablation study, where score is plotted against timesteps. Although the Pong curves look similar, they
are significant based on the standard deviation. We can include additional games in this study for the final paper.

**5.** The oversampling ratio $X$ controls how many extra samples are promoted to the cache during training, which
has implications for the amount of bias and the total runtime. If the cache is very large ($X \gg 1$), then minibatches
drawn from it will be mostly uncorrelated, but a great number of computed $\lambda$-returns would be wasted. It is generally
desirable to use a lower $X$ value for faster runtime when some bias can be tolerated. However, if $X = 1$ (minimal), our
prioritization method could not work because there would be no extra samples to truncate from the cache copies.

Figure 1: DQN($\lambda$) with median-based $\lambda$ selection against a 3-step target-network DQN baseline on six Atari games,
trained for 10 million timesteps. Results were averaged over 5 random seeds and standard deviation is shown.

[Meta-Review · NeurIPS 2019]

The reviewers unanimously support acceptance. We encourage the authors to strongly consider the suggestions provided by the reviewers for improving a camera ready version.